# The Effect of C/Si Ratio and Fluorine Doping on the Gas Permeation Properties of Pendant-Type and Bridged-Type Organosilica Membranes

**DOI:** 10.3390/membranes12100991

**Published:** 2022-10-13

**Authors:** Ikram Rana, Takahiro Nagaoka, Hiroki Nagasawa, Toshinori Tsuru, Masakoto Kanezashi

**Affiliations:** Separation Engineering Laboratory, Department of Chemical Engineering, Hiroshima University, Higashi-Hiroshima 739-8527, Japan

**Keywords:** organosilica membranes, C/Si ratio, fluorine doping, network pore size, gas permeation properties

## Abstract

A series of pendant–type alkoxysilane structures with various carbon numbers (C_1_–C_8_) were used to fabricate sol–gel derived organosilica membranes to evaluate the effects of the C/Si ratio and fluorine doping. Initially, this investigation was focused on the effect that carbon-linking (pendant–type) units exert on a microporous structure and how this affects the gas-permeation properties of pendant–type organosilica membranes. Gas permeation results were compared with those of bridged–type organosilica membranes (C_1_–C_8_). Network pore size evaluation was conducted based on the selectivity of H_2_/N_2_ and the activation energy (*E_p_*) of H_2_ permeation. Consequently, *E_p_* (H_2_) was increased as the C/Si ratio increased from C_1_ to C_8_, which could have been due to the aggregation of pendant side chains that occupied the available micropore channel space and resulted in the reduced pore size. By comparison, these permeation results indicate that pendant–type organosilica membranes showed a somewhat loose network structure in comparison with bridged–type organosilica membranes by following the lower values of activation energies (*E_p_*). Subsequently, we also evaluated the effect that fluorine doping (NH_4_F) exerts on pendant−type [methytriethoxysilane (MTES), propyltrimethoxysilane (PTMS)] and bridged-type [1,2–bis(triethoxysilyl)methane (BTESM) bis(triethoxysilyl)propane (BTESP)] organosilica structures with similar carbon numbers (C_1_ and C_3_). The gas-permeation properties of F–doped pendant network structures revealed values for pore size, H_2_/N_2_ selectivity, and *E_p_* (H_2_) that were comparable to those of pristine organosilica membranes. This could be ascribed to the pendant side chains, which might have hindered the effectiveness of fluorine in pendant–type organosilica structures. The F–doped bridged–type organosilica (BTESM and BTESP) membranes, on the other hand, exhibited a looser network formation as the fluorine concentration increased.

## 1. Introduction

The separation of various gaseous molecules from industrial/natural waste has become a crucial issue. Inherently attractive characteristics, such as low energy consumption and cost effectiveness, have made membrane science the most promising approach to separation by comparison with cryogenic distillation or liquid/solid adsorption [1,2,3]. Various membrane materials (SiO_2_, TiO_2_, ZrO_2_, Zeolite, etc.) have been introduced to develop the microporous thin layers required for different separation applications/systems [4,5,6,7,8,9]. Among all of these materials, amorphous silica (average pore size: 0.34 nm) is considered a potential candidate with high chemical and thermal stability [10]. In the early 1990s, chemical vapor deposition (CVD) and sol–gel methods were employed to fabricate amorphous silica molecular sieve membranes, and both of these processes produced main-chain siloxane (Si–O–Si) and terminal silanol (Si–OH) bonding [11,12,13,14,15,16]. Silica-derived network structures appeared to be smaller (0.34 nm) and were applicable only to helium (He) and hydrogen (H_2_) separation systems, but this type of structure was inappropriate when applied to molecules larger (CO_2_: 0.33 nm and N_2_: 0.36 nm) than helium (He: 0.26 nm) or hydrogen (H_2_: 0.28 nm). Another drawback associated with the traditional silica (TEOS) is its structural destabilization under humid conditions wherein H_2_O molecules interact with membrane surfaces. Consequently, silica network structures undergo densification that results in a drastic decrease in membrane performance [17,18].

To overcome the issues of hydrothermal stability and network pore size tuning of conventional silica, various studies have utilized the incorporation of two commonly known organic molecules, pendant–type alkoxysilane and organic bridged–type alkoxysilane, into inorganic silica (SiO_2_) to fabricate sol–gel derived silica membranes with excellent molecular sieve properties and excellent hydrothermal stability. In general, a typical bridged-type alkoxysilane structure contains organic functional groups between two Si atoms (Si–R–Si, R is the functional group) such as 1,2–bis(triethoxysilyl) methane (BTESM, Si–C_1_–Si), 1,2-bis(triethoxysilyl) ethane (BTESE, Si–C_2_–Si), and bis(triethoxysilyl) propane (BTESP, Si–C_3_–Si), while a pendant–type alkoxysilane network structure consists of organic functional groups directly bonded to the Si atom (R–Si–O_1.5_): methyltriethoxysilane (MTES), phenyltriethoxysilane (PhTES), (trifluoropropyl)triethoxysilane (TFPTES), and ethylenetriethoxysilane (ETES). 

To control the network stability and permeation properties of microporous organosilica (pendant– and bridged–type) membranes, two commonly used “spacer” and “template” methods were employed via an adjustment of the organic chain between two Si atoms (Si–R–Si) and/or the terminal organic (O–Si–R) chain [19,20]. Kanezashi et al. [21] reported a series of organosilica network structures consisting of various carbon numbers (C_1_–C_8_) and concluded that the network pore size regressed using the modified gas-translation model (mG–T), enlarged by an increase in the carbon number between two Si atoms. Despite the enlarged pore sizes of some organosilica structures such as BTESP, bis(trimethoxysilyl)hexane (BTMSH), and bis(triethoxysilyl)octane (BTESO), the permeation properties were not as high as membranes with shorter organic linking units (BTESM, BTESE). Since the increased carbon numbers between two Si atoms occupied enough pore channel space via increased flexibility of a long chain to block the permeation of molecules, these membranes showed a low permeance. In a similar manner, membrane fabricated using pendant–type alkoxysilane, has demonstrated excellent water flux (4 kg m^−2^ h^−1^) and stability over the time period of 18 months. The incorporation of –CH_3_ groups into the silica matrix (SiO_2_) improved the stability of siloxane bonds and reduced the number of hydroxyl groups (OH) [22]. However, microporous analysis of as-prepared pendant–type organosilica network structures demonstrated an inaccessibility to gas (N_2_) due to an aggregation of pendant chains in silica pores, which resulted in a blocking effect [23]. Therefore, pore size controllability has remained elusive since the flexibility of the organic chains in a bridged–type organosilica network structure has proven ineffective for membrane fabrication. Similarly, increasing microporosity of pendant–type network structures through the decomposition of organic groups leads to the formation of additional pores [24]. 

To resolve these drawbacks (network pore size tuning and stability) associated with the conventional silica (TEOS) as well as the organosilica network structures (Si–R–Si), our group proposed an innovative strategy to tune the network pore size by introducing fluoride ions (ammonium fluoride (NH_4_F) into the conventional silica and organosilica matrix. Fluorine–induced membranes have exhibited excellent hydrothermal stability and improved permeation properties. Kanezashi et al. reported the effect of fluorine on conventional silica (TEOS) and bridged organosilica (BTESM) membranes and both membranes showed enlarged network pore size that was affected by Si–F and C–F bonds [25,26]. Fluorine-doped long–chain organosilica membranes with flexible organic linking units (BTESP, Si–C_3_–Si) have demonstrated an enlarged pore size with permeation properties (H_2_ permeance; 10^−6^ mol m^−2^ s^−1^ Pa^−1^) that are at least one order of magnitude higher than those of undoped BTESP membranes (10^−7^ mol m^−2^ s^−1^ Pa^−1^) [27]. To the best of our knowledge, no one has reported the effect of fluoride ions on pendant-type organosilica structures.

In the present study, we chose various mono–silicon pendant-type and bridged–type organosilica based on the carbon number adjacent to the Si atoms for fabrication of the organosilica membranes. Initially, we evaluated network pore sizes based on the gas permeation properties for all pendant groups (C_1_–C_8_), as shown in Appendix A. Next, we evaluated the effect of fluorine (NH_4_F) doping on MTES and PTMS based on the gas-permeation properties. The physiochemical properties of organosilica were examined via XRD, N_2_ adsorption isotherms, and FT–IR. XPS measurement was conducted to observe the fluorine status of pendant–type organosilica structures.

## 2. Experimental Section

### 2.1. F–Doped and Undoped Sol–Gel Preparations

An organosilica sol was prepared via a process of hydrolysis/condensation using the sol–gel method [28,29,30] in an ethanol solution. It should be noted that, all chemicals were kindly supplied by TCI Co., Ltd. Tokyo, Japan. The reaction was catalyzed using nitric acid (HNO_3_) in the preparation of both fluorine-doped and undoped organosilica sols. After the addition of Si into an ethanol solution, the catalyst (HNO_3_) and water (H_2_O) were added dropwise with vigorous stirring at 500 r.p.m. to promote the hydrolysis/condensation at a reaction temperature of 25 °C. The final molar composition of alkoxysilane/water/catalyst was maintained at 1/30/1, and ethanol (EtOH) was utilized to control the 0.5 wt% of Si. To prepare the fluorine–doped pendant/bridged organosilica sols, the fluorine concentration was fixed at 0–50 mol%. Simultaneously, the gels were prepared using a slow drying process at 40 °C under an air atmosphere, which was followed by grinding in a mortar. Gels were calcined at 300–350 °C for structural characterization (XRD, N_2_ adsorption, and XPS analysis).

### 2.2. Characterization of Sol–Gel

Organosilica sols were measured using a DLS analyzer (Zetasizer nano, ZEN3600, Malvern Co., Malvern, UK). A KBr plate was coated dropwise to obtain a measurement on the FT–IR spectrum (FT/IR–4100, JASCO, Tokyo, Japan) within a range of 400–4000 cm^−1^ to evaluate the functional groups in the organosilica network structure. X–Ray diffraction (D2 PHA-SER Bruker, Berlin, Germany) measurement was carried out to analyze the microstructure characteristics of xerogel powders for the organosilica network structures. N_2_ adsorption measurements were conducted using a BELMAX (MicrotracBEL corp., Osaka, Japan). X–ray photoelectron spectroscopy (XPS, Shimadzu, Kyoto, Japan) was used to investigate the fluorine status in the organosilica network structure. To conduct the N_2_ adsorption isotherms and XPS spectra, all samples were evacuated for 12 h at 200 °C prior to starting the measurement.

### 2.3. Fabrication of Organosilica Membranes 

Fluorine–doped and undoped pendant- and bridged–type organosilica membranes were fabricated using a porous alumina (porosity 50%, average pore size; 1µm, length; 100 mm, inner and outer diameter 8–10 mm, respectively) tube supplied by the Nikkato Corporation, Osaka, Japan. First, a porous tube was coated with large (2 µm) and small (0.2 µm) alumina particles diluted with a SiO_2_-ZrO_2_ sol followed by calcination at 550 °C under an air atmosphere. Then an intermediate layer with an average pore size of 1–2 nm was coated with a SiO_2_–ZrO_2_ sol, diluted with distilled water to control the concentration (0.5–1 wt%), and calcined at 550 °C. Finally, organosilica pendant- and bridged-type top separation layers were deposited onto the intermediate layers, which then were calcined at 300–350 °C under a N_2_ atmosphere.

### 2.4. Single–Gas Permeation Measurements

Appendix A depicts the experimental apparatus used for single–gas permeation measurements. Each gas (He (0.26 nm), H_2_ (0.28 nm), CO_2_ (0.33 nm), N_2_ (0.36 nm), CH_4_ (0.38 nm), CF_4_ (0.48 nm), and SF_6_ (0.55 nm)) with high purity was fed from the outer surface of the membrane at 200–400 kPa, and the permeate side was maintained at atmospheric pressure. In the present study, a high level of feed pressure (400 kPa) was applied to the permeation measurement of CF_4_ and SF_6_ molecules for accurate measurement due to the low permeation rate of these molecules. It should be noted that these molecules permeated by Knudsen diffusion, so that permeance was independent of feed pressure. Prior to evaluating the permeation measurements, each membrane was pre–treated at 200 °C under a N_2_ flow for 8 h to remove the adsorbed water. Gas–permeation measurements were carried out at temperatures ranging from 50–200 °C. It should be noted that the experimental deviations of less than 5% were recorded, and all membranes were fabricated 2–3 times to confirm reproducibility. 

## 3. Results and Discussion

### 3.1. Physicochemical Properties of Pendant–Type Organosilica

Figure 1 indicates the FT-IR spectra of various pendant-type organosilica structures coated onto a KBr plate and calcined at 300 °C. The peaks at approximately 1050 cm^−1^ are ascribed to the Si–O–Si bonds irrespective of the pendant-type organosilica structure, and show the completed hydrolysis/condensation reactions during sol–gel preparations [25,26]. Absorption peaks at approximately 1100–1200 cm^−1^ correspond to the Si−C bonds in the organosilica network structures. The absorption peak around 1280 cm^−1^ is associated with the Si–CH_3_ groups and confirm the stability of the methyl groups following calcination at 300 °C. Another peak corresponding to the phenyl group was observed around 1430 cm^−1^ in the PhTES sample [23]. The overall results demonstrate that the chosen calcination temperature is appropriate for membrane fabrication to avoid structural degradation. 

Powder XRD analyses were performed on various pendant–type organosilica gels calcined at 300 °C under a N_2_ atmosphere to confirm the crystalline/microporous structure as shown in Figure 2. The resulting diffraction patterns indicated a prominent broad peak at ~ 2θ = 20°, and this trend is independent of the pendant-type structure (C_1_–C_8_). This broad peak is attributed to the amorphous silica and is likely due to the main organosilica network chain. Meanwhile, the XRD patterns of all samples exhibited a sharp peak at ~ 2θ = 10°, which can be assigned to the well–ordered uniform network structure that could be associated with the pendant organic chain [31]. It should be noted that the peak shift to the slightly lower degree of 2θ = 20° could have been associated with variation in pore size, which will be discussed in the following sections.

Appendix A illustrates the N_2_ adsorption isotherms at 77 K of pendant–type organosilica gels calcined at 300 °C under a N_2_ atmosphere. The pendant-type alkoxysilane showed a negligible amount of adsorbed N_2_ irrespective of the carbon number (C_1_–C_8_), which indicates that pendant–type organosilica possess non–porous structures and a limited number of pores that are accessible to the adsorbed gas. The pendant–type organic chain present in the silica structure is thermally stable at the desired calcination temperature of 300 °C, which might be the reason for it not adsorbing a considerable amount of N_2_. The pendant organic chains acted like barriers among the silica micropores by occupying space in the silica network structure, which would have made the pores relatively inaccessible to N_2_ [23]. It is obvious that the decomposition of organic chains could result in the formation of new micropores that would be accessible to N_2_; this micropore regeneration could only be achieved, however, at the expense of the instability of the network structure [24]. 

Water–contact-angle measurements were carried out to observe the hydrophobic/hydrophilic properties of pendant–type alkoxysilane as shown in Appendix A. All samples showed a water-contact angle that was higher than 80°, which demonstrated the hydrophobicity of the organosilica structures. The contact angle increased as the carbon number increased from C_1_ to C_8_, which further revealed that stable organic chains decrease the silanol (Si–OH) density. In contrast, conventional silica shows a very low water-contact angle due to the excessive amount of silanol groups present in the silica matrix. The successful incorporation of organic moieties increased both network hydrophobicity and thermal stability [23]. 

### 3.2. Pore Size Controllability of Organosilica Membranes

It is quite difficult to obtain an exact pore size distribution of porous membranes for gas separation. The only way to estimate the pore size distribution is to take several measurements of gas permeances as a function of the differences in the molecular sizes of gas molecules. In the present study, the average pore size was roughly estimated according to the molecular size dependence of gas permeances as shown in Appendix A. Figure 3 shows the relationship between H_2_/N_2_ selectivity and the carbon number, and also shows the H_2_ permeance of organosilica membranes at 200 °C. The permselectivity (H_2_/N_2_) of organosilica membranes corresponds to the respective pore size, and clearly decreases with an increase in the carbon number (C_1_ to C_3_). This indicates there is a somewhat loose network formation with an increase in the carbon number, and network pore size slightly decreases as the carbon number in the pendant side chain increases from C_3_ to C_8_. In general, membranes with a larger pore size show a higher level of H_2_ permeance, which can be ascribed to the low resistance against permeate molecules. Even though these membranes were prepared using mono–silicon alkoxysilane that consists of pendant groups with carbon numbers that varied from C_1_ to C_8_, the permeation properties of the membranes showed an approximately similar pore size with a comparable H_2_ permeance even with a higher carbon number attached to the silicon atom (Si). It should be noted that an increase in the H_2_ permeance of pendant-type organosilica network structures occurs at the expense of the removal of methyl groups after firing at an elevated temperature of 550 °C [32].

To further evaluate the network pore size of organosilica membranes, a comparative study was carried out based on the relationship between the C/Si ratio and H_2_/N_2_ permselectivity, as well as the hydrogen activation energy *E_p_* (H_2_). It should be noted that the C/Si ratio of bridged–type membranes is half that of pendant types. Figure 4a features the relationship between pendant- and bridged-type organosilica membranes at 200 °C. In bridged–type organosilica membranes, the permeance ratio of H_2_/N_2_ was largely decreased as the carbon number increased between the two Si atoms, which could have been a result of the formation of a loose network structure [33,34]. On the contrary, pendant-type organosilica membranes showed a somewhat enlarged pore size with permeance ratios of H_2_/N_2_ that were lower than those of bridged–type structures, where higher H_2_/N_2_ selectivity was observed. 

Figure 4b illustrates the relationship between the C/Si ratio and hydrogen activation energy, *E_p_* (H_2_), for pendant- and bridged–type organosilica membranes. Non-adsorptive hydrogen (H_2_) is considered a suitable gas to evaluate pore size based on activation energy (*E_p_*), and H_2_ permeance is less affected by the presence of pinholes. Appendix A shows the temperature dependence of H_2_ permeance for pendant–type (C_1_-C_8_) organosilica membranes calcined at 300 °C under a N_2_ atmosphere. The activation energies (*E_p_*) were obtained using a modified gas–translational model (m–GT) [19,21]. In the bridged–type organosilica network structures, activation energies (*E_p_*) increased as the C/Si ratio increased, which indicates that a smaller pore size required higher activation energy for molecules to diffuse through the micropores. Despite the network pore size of organosilica consisting of higher carbon linking units (BTESP, BTMSH, and BTESO), the flexibility of the organic chains could have affected the permeance of molecules to the point that the main chain could not effectively control the micropore structure. That would have resulted in higher levels of activation energies (*E_p_*) for the membranes with long–chain organic linking units [21]. A similar phenomenon could be in play with the pendant–type organosilica network structure wherein activation energies (*E_p_*) are increased with increases in the C/Si ratio. However, compared with bridged–type organosilica network structures, pendant-type structures show a somewhat lower level of activation energy (*E_p_*), due to the formation of a looser network. 

Figure 5 shows the schematic image of the effect that carbon–linking units exert on the network pore sizes of pendant– and bridged–type organosilica structures. The network pore size of bridged–type organosilica membranes demonstrates a dependence on carbon–linking units; as the carbon-linking units between two Si atoms increased, the network pore size of these membranes increased. An increase in the carbon number resulted in flexible linking units that subsequently led to a failure to maintain microporous properties, and a reduction in permeation properties. Conversely, the pendant–type alkoxysilane structure showed a similar pore size, although the carbon number was higher. A possible reason for the enlarged network pores in the pendant alkoxide structure could have been the existence of hydrophobic pendant side chains, which could have aggregated in the pore wall, and would have resulted in an enlarged network pore size. 

### 3.3. The Effect Fluorine Doping Exerts on A Pendant–Type Organosilica Network Structure (C_1_, C_3_)

Figure 6 shows the FT-IR spectra of fluorine-doped (F/Si = 1/9) and undoped pendant–type (MTES and PTMS) organosilica network structures before/after calcination at 300 °C. A Si–O–Si peak around 1100 cm^−1^ was observed in all samples irrespective of the fluorine doping, which indicates the completion of hydrolysis/condensation reactions during the sol–gel process. Another peak centered around 900 cm^−1^ is ascribed to the silanol groups (Si–OH) present in the sample. However, no apparent difference was observed in the absorption peaks of silanol groups (Si–OH) in either fluorine–doped or undoped samples. Appendix A summarizes the Si–OH/Si–O–Si peak-area ratios of fluorine– doped and undoped samples before/after calcination. The results indicating a slight decrease in silanol groups (Si–OH) after fluorine doping were obtained for both forms of pendant–type organosilica network structures. It should be noted that various studies have reported that fluorine significantly decreases the silanol density (Si–OH) of conventional silica (TEOS) and bridged–type organosilica (BTESM, BTESP) network structures [25,26,27]. This effectiveness of fluorine is associated with a decrease in the silanol density (Si-OH) perturbed by the presence of Si–F and C–F groups in the bridged–type organosilica network structures [35]. However, the present status of fluorine effectiveness in the pendant–type organosilica network structure seemed to be lessened by the existence of pendant–chain aggregation, which might have restricted the chemical bond formation of fluoride ions with silicon (Si–F) and carbon (C–F), as reported in fluorine–doped silica network structures [26].

Appendix A depicts the XRD patterns of fluorine-doped (F/Si = 1/9) and undoped pendant–type organosilica gels calcined at 300 °C under a N_2_ atmosphere. The XRD pattern of organosilica gels exhibited two peaks at ~ 2θ = 20° and at ~ 2θ = 9°, respectively. The first peak corresponds to the amorphous structure of the main siloxane chain (Si–O–Si) in all samples irrespective of the doped fluorine, and the second sharp peak (~ 2θ = 9°) is ascribed to the formation of a highly ordered organosilica network structure due to the existence of pendant organic groups. No significant effect was observed in fluorine–doped samples in terms of the peak shift, although previous studies have reported that fluorine–doped samples showed a peak shift to the lower degree of 2θ, which revealed an enlarged Si–O–Si bond angle caused by the Si–F groups present in the fluorine–doped organosilica network structures [25,26]. These results are consistent with the FT–IR spectra, wherein no obvious change in peak shift (blue shift) was observed after fluorine doping into pendant–type organosilica structures. 

Microporous properties were evaluated via N_2_ adsorption isotherms for fluorine–doped (F/Si = 1/9) and undoped pendant–type organosilica gels calcined at 300 °C, as shown in Figure 7. A negligible amount of N_2_ adsorption was observed in undoped samples, which indicated that most of the pores were inaccessible and corresponded to a non–porous network structure. N_2_ adsorption slightly improved in fluorine–doped samples, which suggests that micropores were generated after fluorine incorporation into the pendant–type organosilica network structure. A possible reason for the improved adsorption properties of F–doped samples is that the reduced silanol density (Si–OH) was affected by the fluoride ions. A similar effect of increased microporosity after fluorine doping was observed in the non–porous BTESP network structure. A high uptake of N_2_ adsorption was observed in fluorine-doped BTESP samples, whereas undoped BTESP samples showed a non-porous structure. This change in the microporous properties of F–BTESP samples is associated with a decrease in silanol density (Si–OH), which was perturbed by Si–F and C–F bonds and the subsequent network structure resulted in a high surface area and micropore volume [27]. Thus far, these results are consistent with the observation of physicochemical properties (FT–IR), where a small change in silanol density (Si–OH) was seen in fluorine-induced pendant–type organosilica samples, and F–doped gels simultaneously showed improved N_2_ adsorption properties. 

Figure 8 shows H_2_O adsorption isotherms at 25 °C for fluorine–doped (F/Si = 1/9) and undoped pendant-type organosilica gels calcined at 300 °C under a N_2_ atmosphere. Prepared samples (MTES and PTMS) showed H_2_O adsorption onto pendant–type organosilica surfaces. Conversely, fluorine-doped gels displayed a negligible amount of H_2_O adsorption, which is an indication of an increase in the hydrophobicity of fluorine–doped gels. The incorporation of fluorine into hydrophobic pendant–type organosilica further improved the hydrophobic/hydrophilic properties. Several studies have reported similar results of increased hydrophobicity with the addition of fluoride ions into silica matrix by showing the very low amount of H_2_O adsorbed by comparison with undoped samples [26,36,37].

Appendix A shows the XPS spectra ranging from 0–1200 eV for fluorine-doped (F/Si = 1/9) and undoped pendant–type organosilica gels calcined at 300 °C under a N_2_ atmosphere. Organosilica–derived gels demonstrated Si peaks (100 and 150 eV), a C 1s peak (280 eV), and an O 1s peak (520 eV) irrespective of fluorine doping. An F 1s peak around 690 eV was detected only in fluorine–doped gels, which confirms the existence of fluorine in the F–doped pendant-type organosilica gels.

Appendix A represents the narrow spectra of the F 1s peak, which ranged from 695 to 680 eV for both fluorine-doped and undoped pendant–type organosilica gels calcined at 300 °C. As–prepared samples showed no peak intensity indicating Si–F bonds. On the other hand, a peak was detected as Si–F (688 eV) in all fluorine–containing organosilica network structures. The existence of fluorine as Si–F and C–F bonds has also been reported in bridged organosilica (BTESM, Si–C_1_–Si; and BTESP, Si–C_3_–Si) network structures [26,27].

### 3.4. Network Pore-Size Evaluation of Pendant– and Bridged–Type Organosilica Membranes

The molecular size dependence and temperature dependence of H_2_ permeance for fluorine–doped MTES and PTMS pendant–type (C_1_–C_3_) organosilica membranes is shown in Appendix A, respectively. Figure 9 shows the relationship between fluorine concentration (F/Si) and gas permeation properties (H_2_ permeance, H_2_/N_2_, and *E_p_* of H_2_) at 200 °C for fluorine-doped and undoped pendant–type (a) and bridged–type (b) organosilica membranes calcined at 300 °C. Approximately similar H_2_ permeance values were observed for both pendant–type organosilica membranes irrespective of the carbon number (C_1_ and C_3_). After fluorine incorporation (0–50 mol %), permeation properties were not significantly improved, and permselectivity corresponding to the network pore size remained unchanged, as well as the *E_p_* (H_2_). These results indicate that the network pore size of both pendant–type membranes (C_1_ and C_3_) was almost independent of fluorine concentration. 

On the other hand, bridged–type organosilica membranes showed a strong dependence on the carbon number present between two Si atoms. The effect of carbon linking units was apparent when the BTESM (Si–C_1_–Si) with a single carbon showed a higher level of H_2_ permeance compared with that of BTESP (Si–C_3_–Si), with three carbons between two silicon atoms. Although previous studies have reported that permeation properties were improved as the carbon number increased, the flexibility of organic linking units could have decreased the H_2_ permeance in BTESP membranes [21,37]. Long–chain organosilica membranes are known to block pores due to space occupation, and these are not considered effective for the fabrication of highly permeable gas–separation membranes. After fluorine (NH_4_F) incorporation, improved permeation properties have been reported irrespective of the organic linking units, which demonstrated an enlarged network pore size that was caused by Si–F and C–F bonds [27,35]. H_2_/N_2_ selectivity and *E_p_* (H_2_) decreased as the fluorine concentration increased (F/Si = 0–5/5), which indicates a loose network formation. Network pore sizes of organic/inorganic silica membranes have been controlled effectively with appropriate fluorine concentration. These results further proved that the significance of fluorine in bridged–type organosilica is independent of the Si precursor.

Figure 10 shows the schematic illustration of fluorine–doped organosilica pendant–type and bridged-type network structures. The catalytic effect of fluoride ions is the progression of hydrolysis/condensation reactions, as well as a reduction in the silanol density (Si–OH). Herein, bridged organosilica structures showed a cleavage of Si–C [38,39,40,41] bonds, which later formed C–F; simultaneously, this hydrophobic bond interaction reduced the OH groups. This dissociation energy of Si–C bonds could have resulted in cleavage and the subsequent formation of hydrophobic C–F bonds. This chemical bond formation affected the desired pore size and improved the permeation properties of bridged–type organosilica membranes. On the contrary, no change in network pore size was observed in fluorine–doped pendant–type organosilica structures. The possible reason for the fluorine ineffectiveness could be explained by the aggregation of pendant organic groups in the micropores, which might have restricted the fluorine bond formation to Si–C and C–F. 

## 4. Conclusions

A series of pendant alkoxysilane structures with various carbon numbers (C1–C8) were used to fabricate sol–gel derived organosilica membranes to evaluate the effects of the C/Si ratio and fluorine doping. Initially, this investigation was focused on the effect that carbon-linking (pendant–type) units exert on a microporous structure and how this affects the gas–permeation properties of pendant–type organosilica membranes. Gas permeation results were compared with those of bridged–type organosilica membranes (C1–C8). Subsequently, we also evaluated the effect that fluorine doping (NH4F) exerts on pendant–type [methytriethoxysilane (MTES), propyltrimethoxysilane (PTMS)] and bridged–type [1,2–bis(triethoxysilyl)methane (BTESM) bis(triethoxysilyl)propane (BTESP)] organosilica structures with similar carbon numbers (C1 and C3). 

The pendant-type organosilica membranes showed a slightly looser network structure with a decrease in H_2_/N_2_ selectivity compared to those of bridge-type organosilica membranes. Activation energy (*E_p_*) increased as the carbon number increased in both types of membranes, which indicates increased space occupation by pendant side chains as the C/Si ratio increased, and increased network flexibility in bridged–type organosilica membranes. However, permeation results revealed that pendant–type organosilica membranes demonstrated a larger pore size in comparison with bridged–type organosilica membranes, due to the lower values of activation energies (*E_p_*) which is evidence of a loose network formation.

Fluorine–doped pendant–type organosilica membranes showed pore sizes similar to those of undoped organosilica membranes with comparable selectivity (H_2_/N_2_) and *E_p_* (H_2_). Those results are ascribed to the existence of pendant side chains, which could have restricted the effectiveness of fluorine and allowed the pore size to remain unchanged. Conversely, fluorine significantly improved the permeation properties of bridged–type organosilica membranes (H_2_/N_2_); the F/Si ratio was increased due to the formation of a loose network formation caused by the Si–F and C–F groups present in the network structure. 

After several experiments, all membranes continued to exhibit considerable stability in characteristics such as molecular size dependence and temperature dependence of gas permeance. These good results notwithstanding, long-term separation performance of these membranes will be conducted in our future work.

## Figures and Tables

**Figure 1 membranes-12-00991-f001:**
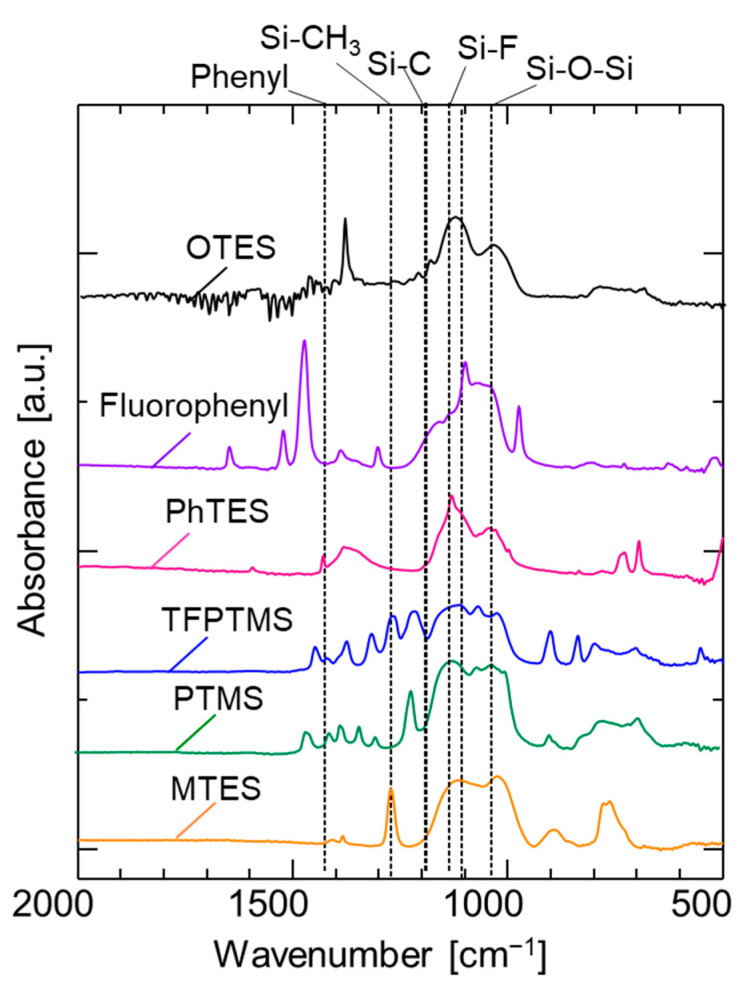
FT–IR spectra of various pendant–type organosilica films calcined at 300 °C under a N_2_ atmosphere.

**Figure 2 membranes-12-00991-f002:**
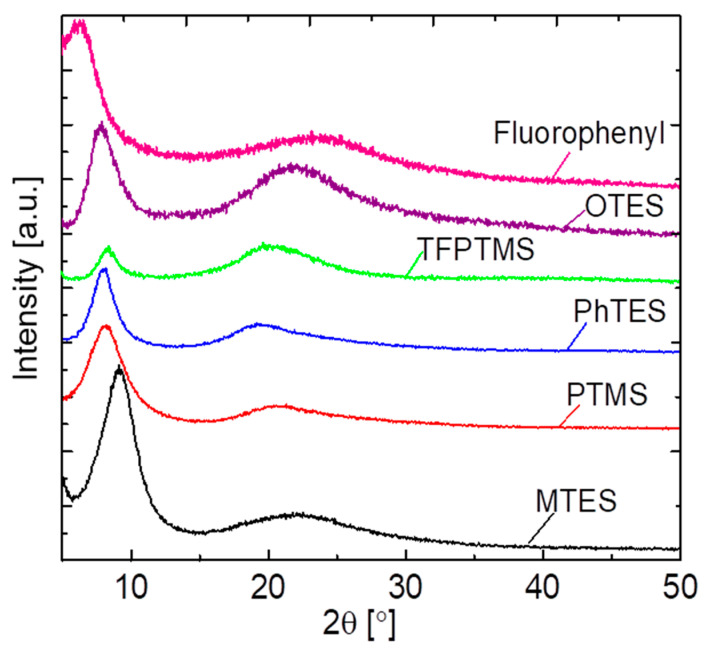
XRD patterns of various pendant–type silica–derived gels calcined at 300 °C under a N_2_ atmosphere.

**Figure 3 membranes-12-00991-f003:**
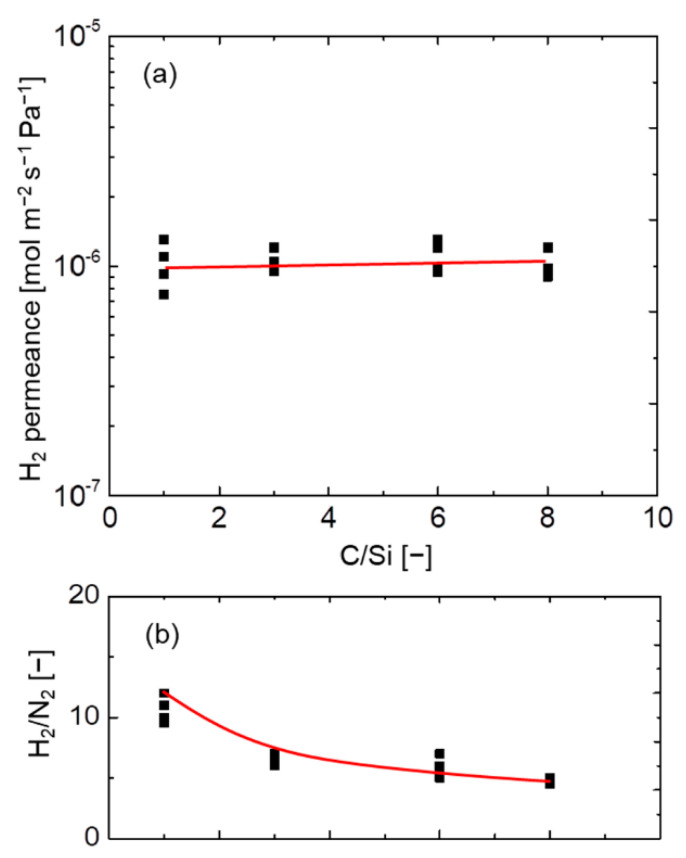
Relationship between the C/Si ratio and (**a**) H_2_ permeance and (**b**) H_2_/N_2_ at 200 °C for pendant–type organosilica membranes calcined at 300 °C under a N_2_ atmosphere.

**Figure 4 membranes-12-00991-f004:**
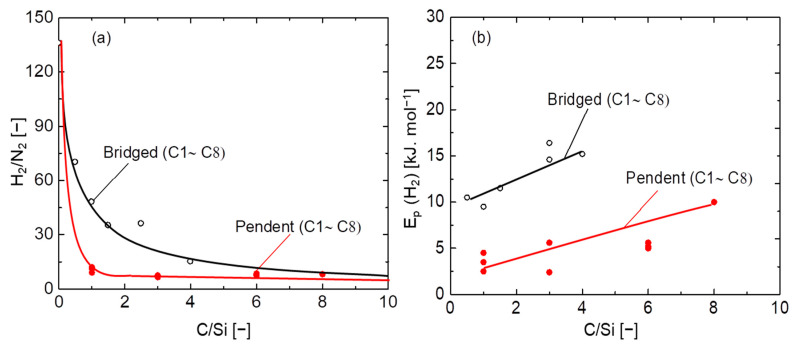
Relationship between C/Si ratios, H_2_/N_2_ selectivity at 200 °C (**a**), and activation energy (**b**) for pendant– and bridged–type organosilica membranes calcined at 300–350 °C under a N_2_ atmosphere.

**Figure 5 membranes-12-00991-f005:**
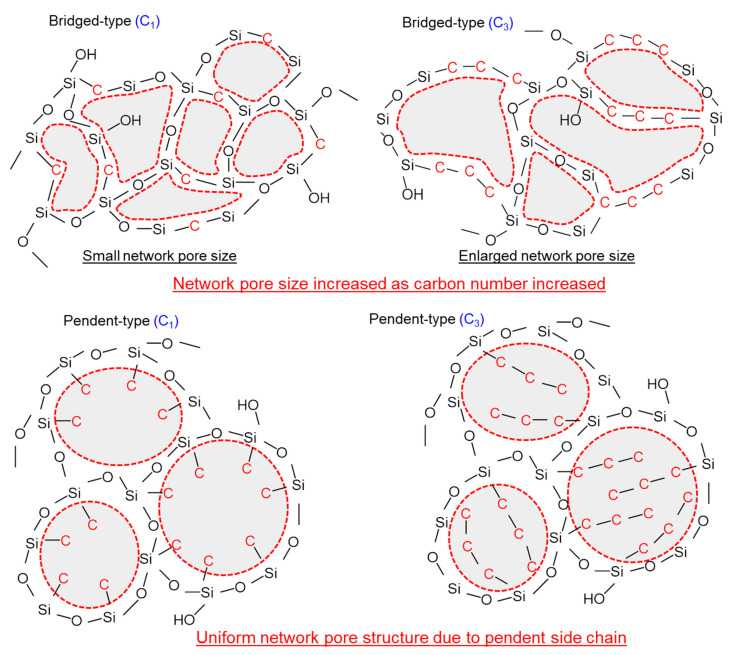
Effect that C/Si ratio exerts on pendant-type and bridged–type organosilica network pore sizes.

**Figure 6 membranes-12-00991-f006:**
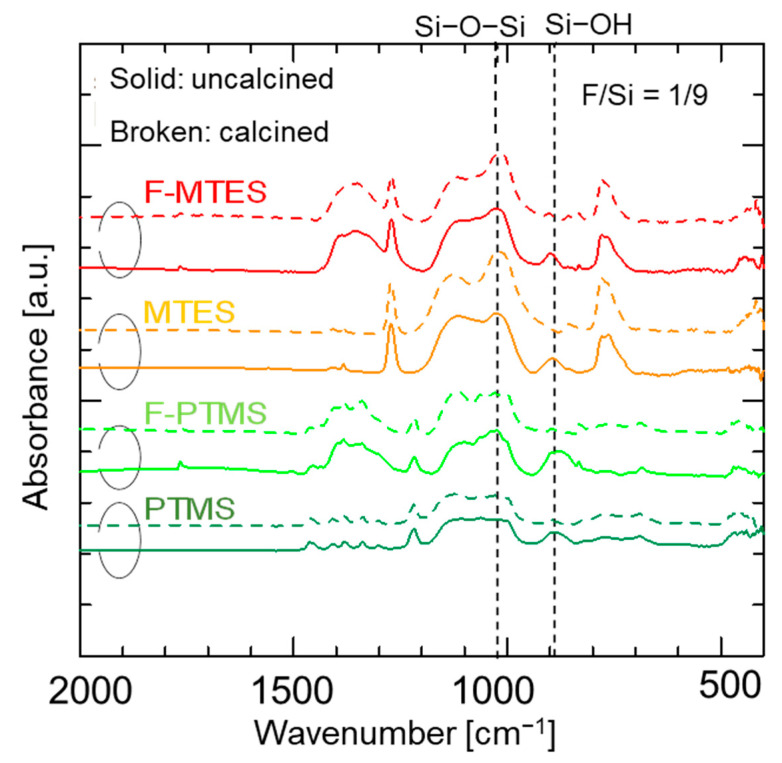
FT–IR spectra of fluorine–doped (F–MTES, F–PTMS) and undoped (MTES, PTMS) pendant–type organosilica films before/after calcination at 300 °C under a N_2_ atmosphere.

**Figure 7 membranes-12-00991-f007:**
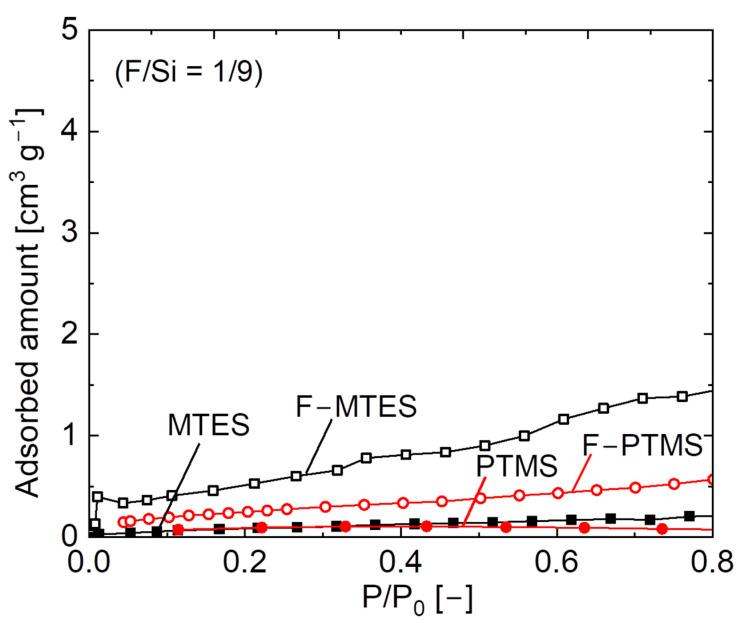
N_2_ adsorption isotherms of fluorine–doped (F–MTES, F–PTMS) and undoped (MTES, PTMS) pendant–type organosilica gels calcined at 300 °C under a N_2_ atmosphere.

**Figure 8 membranes-12-00991-f008:**
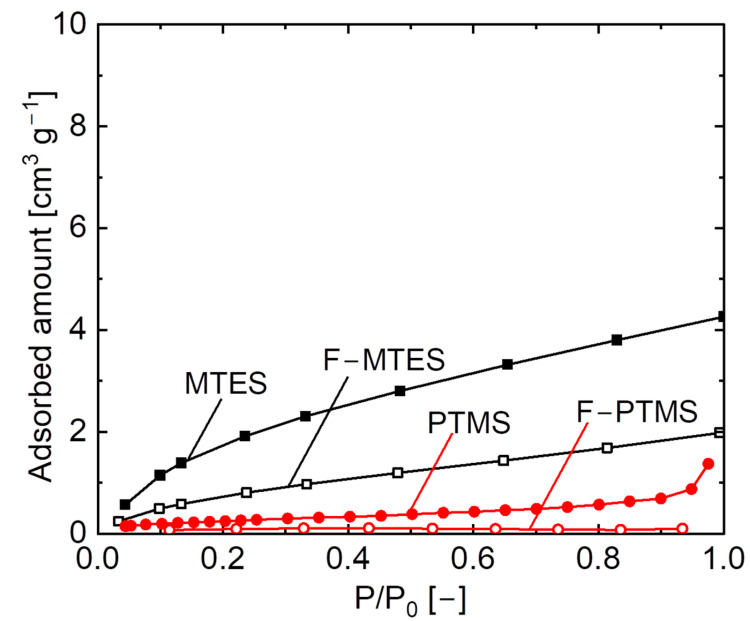
H_2_O adsorption isotherms at 25 °C for fluorine-doped (F–MTES, F–PTMS) and undoped (MTES, PTMS) pendant–type organosilica gels calcined at 300 °C under a N_2_ atmosphere.

**Figure 9 membranes-12-00991-f009:**
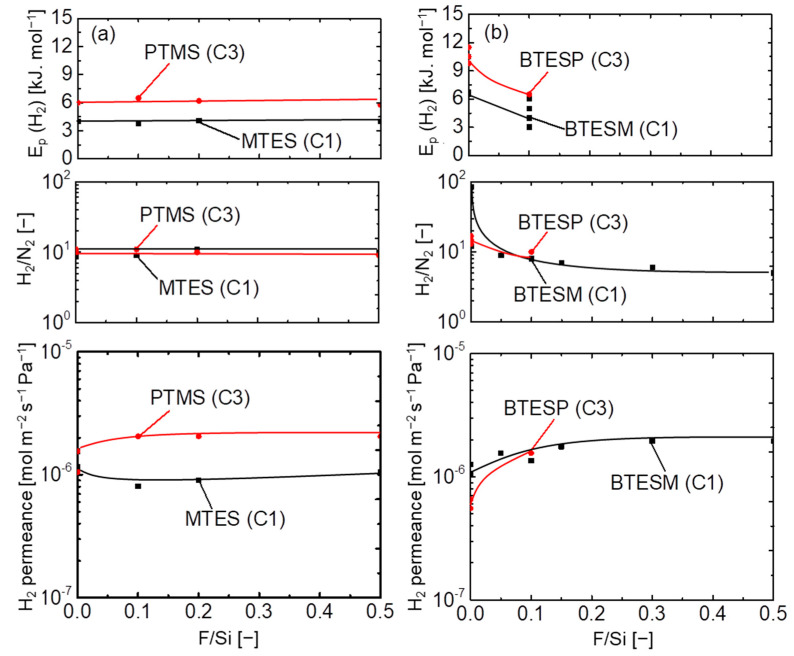
Effect of doped fluorine concentration on the network pore sizes of pendant– (**a**) and bridged–type (**b**) organosilica membranes calcined at 300–350 °C under a N_2_ atmosphere.

**Figure 10 membranes-12-00991-f010:**
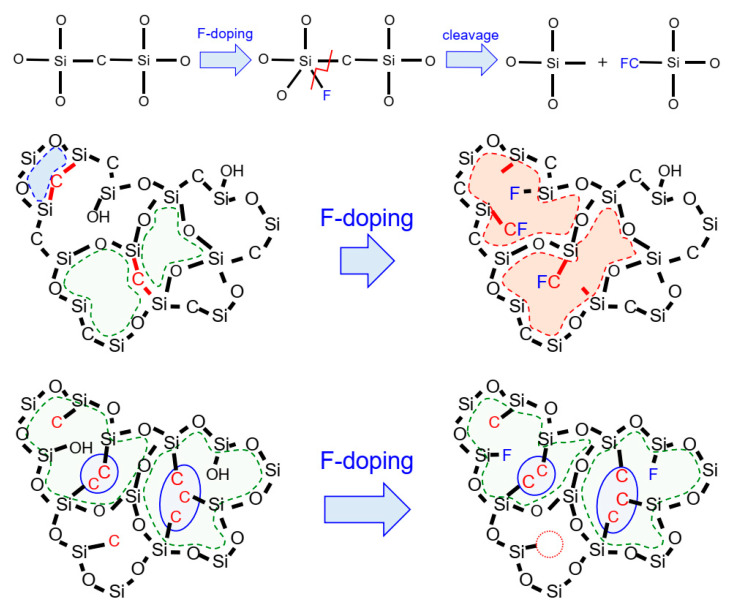
A schematic image of the effect that fluorine doping exerts on pendant–type and bridged–type organosilica network structures calcined at 300–350 °C under a N_2_ atmosphere.

## Data Availability

The data presented in this study are available on request from the corresponding author.

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
