# Peer review of "The Effect of C/Si Ratio and Fluorine Doping on the Gas Permeation Properties of Pendant-Type and Bridged-Type Organosilica Membranes"

_membranes, 2022, doi:10.3390/membranes12100991_

Round 1

Reviewer 1 Report

In this manuscript, the authors presented the preparation of different sol-gel-derived organosilica membranes by employing a series of mono-silicon pendant- and bridged-type organosilicas based on the carbon number (C1-C8) adjacent to Si atoms. The effect of C/Si ratio and fluorine doping on the physiochemical properties, network pore sizes, and gas permeation properties of organosilica membranes were systematically investigated; meanwhile, the plausible influencing mechanisms were put forward. The workload of this manuscript is substantial and some interesting phenomena and data have been found in the experimental results. However, taking consideration of some mistakes or doubts in the manuscript, I recommend this for publication after a major revision based on the following questions and comments.

1.    In Figure 4 and 9, the variation trends of nitrogen adsorbed amount against P/P0 for various pendant-type silica-derived gels were not obvious. Please modify the figure with adequate y-axis value.

2.    In figure 3, for various pendant-type silica-derived gels, the 2θ value almost decreased with the increasing carbon number, why? Is it an indication of the pore size variation?

3.    For “3.1. Physicochemical properties of pendant-type organosilica line191-198”, the water-contact angle images for pendant-type alkoxysilane should be provided.

4.    As described in the manuscript, the activation energy (Ep) is an important parameter for reflecting the variation of pore size, the specific calculation process of Ep for at least one example should be provided. In addition, why is the higher Ep value the indication of a smaller pore size?

5.    Why are hydrogen molecules chosen for calculating the activation energy? How about other gas molecules like He or N2?

6.    In line 218-220, a grammatical mistake was found in the sentence, reorganize the sentence.

7.    Other minor remarks.

For reference 1, Volume and Pages were missed

For reference 11, right Journal Abbreviation should be Science

For reference 15 29 35, the dot before Year should be deleted

For reference 32, right journal abbreviation should be Int. J. Hydrog. Energy

Reviewer 2 Report

The manuscript summarizes the effect if fluorine doping and C/Si ratio on the gas separation performances of pendant-type and bridged-type organosilica membranes. Although there have been many reports dealing with the similar materials, the report here is useful for readers working on the corresponding fields. I think the manuscript is informative for the various readers, and can be accepted for publication after revision. Several comments are given below, which could be improving the manuscript.

1-      The introduction section should be reorganized. Pay attention on the most important aspects related to this topic and provide a clear presentation of the state of art in this field.

2-      The novelty of this study should be clearly highlighted. Abstract and conclusion do not reflect any novelty of the work.

3-      The authors should take care of some grammatical errors and spacing throughout the paper. Some mistyping in the text should be corrected.

4-      The data points (experimental gas separation data) require standard deviation.

5-      What instrument was used for analysis of the synthesized membranes? Please add in Section 2.

6-      Gas separation with different membranes is complex. There is a lot of parameters that determine its efficiency. In the manuscript I cannot find information about the influence of temperature, pressure, etc. on the separation performances of the fabricated membranes. Why these parameters weren't analyzed? In order to optimize the separation process detailed description of influence of each of those parameters should be presented.

7-      To determine the applicability of the fabricated membranes, the long-term gas separation test would be essential. Please do this test.

8-      There are many images (12 images) in this manuscript. Please move some unnecessary images to the supplementary file.

Reviewer 3 Report

The manuscript investigated the effect of C/Si ratio and fluorine doping on organosilica membranes. I recommend to accept this paper after addressing the follow questions.

(1)  The reference to Figure 3 is missing in line 165.

(2)  Figure 4 and Figure 9 should be modified: It is hard to read the data.

(3)  2.1 heading is the same as 2.2 heading, please revise.

(4)  The influence of temperature and pressure on single-gas permeation was mentioned in 2.4. However, the raw data and description was missed from the main text. Also the single gas permeance of CO2, CH4, CF4 and SF6 were missed.

(5)  Please explain why the wavenumber of phenyl group was different (line 159 vs Figure 2).

(6)  NH4F was used as the fluorine source. However, it is hard for me to understand how the C-F bond formed. Please give more explanation or evidence regarding Figure 12.

Round 2

Reviewer 1 Report

The authors have fully addressed all the concerns raised. It is therefore recommened for acceptance in the current form.

Reviewer 2 Report

Accept in present form